# New Insights on *Acanthus ebracteatus* Vahl: UPLC-ESI-QTOF-MS Profile, Antioxidant, Antimicrobial and Anticancer Activities

**DOI:** 10.3390/molecules27061981

**Published:** 2022-03-18

**Authors:** Opeyemi Joshua Olatunji, Oladipupo Odunayo Olatunde, Titilope John Jayeoye, Sudarshan Singh, Sirinporn Nalinbenjapun, Sasikarn Sripetthong, Warangkana Chunglok, Chitchamai Ovatlarnporn

**Affiliations:** 1Traditional Thai Medical Research and Innovation Center, Faculty of Traditional Thai Medicine, Prince of Songkla University, Hat Yai 90110, Songkhla, Thailand; 2Department of Food and Human Nutritional Sciences, Faculty of Agricultural and Food Sciences, University of Manitoba, Winnipeg, MB R3T 2N2, Canada; oladipupo.olatunde@umanitoba.ca; 3Department of Chemistry, Faculty of Physical Science, Alex-Ekwueme Federal University Ndufu-Alike, Abakalilki P.M.B 1010, Ebonyi State, Nigeria; titilope12@gmail.com; 4Food Technology and Innovation Research Center of Excellence, Institute of Research and Innovation, Walailak University, Nakhon Si Thammarat 80160, Nakhon Si Thammarat, Thailand; sudarshansingh83@hotmail.com (S.S.); cwarang@wu.ac.th (W.C.); 5Department of Pharmaceutical Chemistry, Faculty of Pharmaceutical Sciences, Prince of Songkla University, Hat Yai 90112, Songkhla, Thailand; nasirin109@hotmail.com (S.N.); sasikarn.spt@gmail.com (S.S.); 6Drug Delivery System Excellent Center, Faculty of Pharmaceutical Sciences, Prince of Songkla University, Hat Yai 90112, Songkhla, Thailand; 7School of Allied Health Sciences, Walailak University, Nakhon Si Thammarat 80160, Nakhon Si Thammarat, Thailand

**Keywords:** *Acanthus ebracteatus* Vahl, antioxidant, antibacterial, anticancer, polyphenols

## Abstract

This study investigated the antioxidant, antimicrobial, anticancer, and phytochemical profiling of extracts from the leaves and stem/root of *Acanthus ebracteatus* (AE). The total phenolic content (TPC), total flavonoid content (TFC), 2,2-diphenyl-1-picryl-hydrazyl-hydrate (DPPH) radical-scavenging activity, 2, 2′-azino-Bis-3-ethylbenzothiazoline-6-sulfonic acid (ABTS) radical-scavenging activity, metal chelating activities (MCA), ferric reducing antioxidant power (FRAP) and oxygen radical antioxidant capacity (ORAC) were used for antioxidant assessment. The ethanolic extracts of the leaves (AEL-nor) and stem/root (AEWP-nor) without chlorophyll removal and those with chlorophyll removal, using sedimentation process (AEL-sed and AEWP-sed), were prepared. Generally, AEL-sed showed the highest antioxidant activity (FRAP: 1113.2 µmol TE/g; ORAC: 11.52 µmol TE/g; MCA: 47.83 µmol EDTA/g; ABTS 67.73 µmol TE/g; DPPH 498.8 µmol TE/g; TPC: 140.50 mg/GAE g and TFC: 110.40 mg/CE g) compared with other extracts. Likewise, AEL-sed also showed the highest bacteriostatic (MIC) and bactericidal (MBC) effects, as well as the highest anticancer and antiproliferative activity against oral squamous carcinoma (CLS-354/WT) cells. UPLC-ESI-QTOF/MS analysis of AEL-sed and AEWP-sed tentatively identified several bioactive compounds in the extracts, including flavonoids, phenols, iridoids, and nucleosides. Our results provide a potentially valuable application for *A. ebracteatus*, especially in further exploration of the plant in oxidative stress-related disorders, as well as the application of the plant as potential nutraceuticals and cosmeceuticals.

## 1. Introduction

Medicinal plants have played a pivotal role in primary health care over the past few decades, especially in low- and middle-income countries. Several medicinal plants have been the building blocks for the successful discovery of bioactive medicinal agents currently used in the treatment of a wide range of diseases. Furthermore, medicinal plants are generally perceived as safer substituents for the treatment of devastating diseases, including diabetes, cancer, cardiovascular disorders, and neurodegenerative diseases amongst others [1,2,3]. The roles of these natural endowments in oxidative stress-related diseases have been extensively explored. The display of excellent antioxidant activities by plant extracts is directly correlated to the existence of bioactive constituents, notably polyphenolic compounds, which make these medicinal plant extracts display properties indicating they are significant antidiabetic, anti-inflammatory, antiaging, and anticancer agents [3,4].

*Acanthus ebracteatus* Vahl. (Sea Holly) is a multipurpose mangrove medicinal plant belonging to the Acanthaceae family that grows in several southeast Asian countries including Thailand, Malaysia, Indonesia, the Philippines, and Vietnam [5,6]. *A. ebracteatus* has several traditional folk medicinal uses, especially in the treatment of rheumatism, cough, snake-bite, chronic fever, asthma, hepatitis, intestinal worms, preventing hair loss, herpes zoster, leucorrhea, wound, menstrual disorders, rash, and skin diseases [5]. Although there are few reports regarding the phytochemical richness of *A. ebracteatus*, previous studies have reported the presence of aliphatic alcohol, aliphatic glycosides, phenolic glycosides, terpenes, megastigmane glycosides, flavonoids, and lignan glycosides [6,7]. *A. ebracteatus* has been pharmacologically reported to show anti-inflammatory, neuroprotection and wound healing effects [5,6,8,9]. However, none of these reports provided detailed information regarding the phytochemical and pharmacological profiles of the leaves and the stem/root extracts of this species. As such, this work evaluated the chemical composition, antioxidant, antibacterial and cytotoxic activities of *A. ebracteatus* leaves and stem/root extracts.

## 2. Results

### 2.1. Evaluation of the Antioxidant Activity

The antioxidant activities of the leaves and stem/root extracts of *A. ebracteatus* prepared without chlorophyll removal and with the sedimentation chlorophyll removal method described in Section 4.2 were examined using several in vitro assays (DPPH, ABTS, FRAP, MCA, and ORAC). It was observed from the results that the leaves extract (AEL) showed better antioxidant activity in all the assays compared to the extract from the stem/root (AEWP). The leaves extract from the sedimentation process (AEL-sed) showed the highest antioxidant activity in the FRAP (1113.2 ± 4.2 µmol TE/g), ORAC (11.52 ± 0.3 µmol TE/g), MCA (47.83 ± 0.01 µmol EDTA/g), ABTS (67.73 ± 0.5 µmol TE/g) and DPPH (498.8 ± 0.4 µmol TE/g) assays (Table 1).

### 2.2. Total Phenolic and Flavonoid Content

Likewise, in the total phenolic (TFC) and total flavonoid content (TFC) quantification, AEL extract also showed higher phenolic and flavonoid contents (AEL-sed; TFC: 110.4 ± 0.5 mg CE/g; TPC: 140.5 ± 0.1 mg GAE/g; AEL-nor; TFC: 107.6 ± 0.02 mg CE/g; TPC: 138.2 ± 0.1 mg GAE/g) compared to AEWP (AEL-sed; TFC: 28.82 ± 0.1 mg CE/g; TPC: 136.88 ± 0.1 mg GAE/g; AEL-nor; TFC: 20.24 ± 0.2 mg CE/g; TPC: 30.49 ± 0.1 mg GAE/g) (Table 1).

### 2.3. Evaluation of Antimicrobial Activity

The antimicrobial activity (MIC and MBC) of the extracts against *E. coli and L. monocytogenes* is shown in Table 2. The results showed that the AEL extract displayed more potent antibacterial activity compared to AEWP, with MIC values ranging from 0.25–0.5 mg/mL and MBC values of 0.5–1.0 mg/mL (Table 2).

### 2.4. Evaluation of Anticancer Activity

The effects of the extracts on the cellular viability of CLS-354 / WT cells were tested following a tetrazolium-based MTT assay. The results demonstrated that AEL-sed and AEWP-sed resulted in a significant (*p* < 0.001) reduction in cell viability of CLS-354/WT in a dose-dependent manner. The ED_50_ was 200 μg/mL and 400 μg/mL for AEL-sed and AEWP-sed, respectively (Figure 1).

### 2.5. Evaluation of the Anti-Proliferative Effect

The AEL-sed and AEWP-sed extracts were further investigated for their anti-proliferative effects on CLS-354/WT. As shown in Figure 2, The results indicated that both extracts significantly (*p* < 0.001) inhibited the proliferation of cells. Moreover, the extracts also retarded cell migration. However, cell viability was not significantly affected. These results suggested a pronounced anti-proliferative effect of AEL-sed and AEWP-sed extract on CLS-354/WT at the tested concentration.

### 2.6. Identification of Compounds in AEL-sed and AEPW-sed

The profiling of the phytochemical constituents in AEL-sed and AEWP-sed was performed via UPLC-QTOF-MS analysis, in the negative ionization mode. The chromatogram of the peaks of the eluted compounds in AEL-sed and AEPW-sed showed several peaks (within 20 min), suggesting the presence of several constituents in the extract (Figure 3). The compounds with mass error <5 ppm and high relative abundance are presented in Table 3 and Table 4. The data shown in Table 3 indicated that the majority of the compounds tentatively identified in AEL-sed were glycosidic constituents, especially flavonoids and phenolic compounds. According to Table 3, simple phenolic and phenolic glycosides, including caffeic acid (Rt: 6.36  min), dihydroferulic acid 4-O-glucuronide (Rt: 5.52 min), 4-glucogallic acid (Rt: 5.532 min), chlorogenic acid (Rt: 7.038 min), kelampayoside A (Rt: 5.269 min) and hydrojuglone glucoside (Rt: 8.018 min) were tentatively identified in AEL-sed.

The several flavonoids and flavonoid glycosides tentatively identified in AEL-sed included luteolin (Rt: 10.478 min), apigenin (Rt: 11.281 min), diosmetin (Rt: 11.432 min), saponarin (Rt: 7.114 min), scutellarein 7-glucuronosyl-(1->2)-glucuronide (Rt: 7.415 min), luteolin 3′-methyl ether 7,4′-dixyloside (Rt: 7.54 min), genistein 4′,7-O-diglucuronide (Rt: 7.917 min), baicalin (Rt: 8.884 min), diosmetin 7-O-beta-D-glucuronopyranoside (Rt: 8.971 min), tetramethylquercetin 3-rutinoside (Rt: 9.147 min) and vestitone 7-glucoside (Rt: 8.386 min). Other flavonoids, including 4′-O-methylglucoliquiritigenin, luteolin 7,3′-dimethyl ether 5-glucoside, and apigenin 7-(3″-acetyl-6″-E-p-coumaroylglucoside were also identified in AEL-sed. Aside from the polyphenolic compounds identified in AEL-sed, the UPLC-QTOF-MS data revealed the presence of several other classes of compounds, such as nucleosides (adenine guanosine and 8-hydroxyadenine), alkaloids (calystegine B5, evoxanthidine and dihydrocapsiate), coumarins (aesculin, esculetin, and osmanthuside B), and quassinoids (sergeolide).

Table 4 summarizes the tentative chemical constituents identified in AEWP-sed. Likewise, the retention times, absorbance spectra, and the data of MS were used to determine the tentative chemical composition of AEWP-sed. The results also suggested that AEWP possesses several classes of compounds, including alkaloids, terpenoids, flavonoids, and phenols, similarly to AEL-sed. The phytochemical profile of AEWP-sed looked similar to that of AEL-sed. However, the compounds such as quinic acid (Rt: 2.302), citbismine C (Rt: 2.653 min), 3′-glucosyl-2′,4′,6′-trihydroxyacetophenone (Rt: 5.904 min), geniposide (Rt: 6.971 min), 7-hydroxy-4-methylphthalide O-[arabinosyl-(1->6)-glucoside] (Rt: 7.047 min), echinacoside (Rt: 7.272 min), dihydromelilotoside (Rt: 7.649 min), (+)-lyoniresinol 9-glucoside (Rt: 7.787 min), prupaside (Rt: 8.377 min), larycitrin 3-(4″-malonylrhamnoside) (Rt: 9.055 min), 2′-hydroxygenistein 7-(6″-malonylglucoside) (Rt: 9.18 min), gomphrenol 3-methylether 4′-glucuronide (Rt: 9.381 min), trans-grandmarin (Rt: 9.934 min), isopetasoside (Rt: 10.197 min), amorphigenol O-vicianoside (Rt: 10.586 min), alpha-peroxyachifolide (Rt: 12.569 min), mollicellin B (Rt: 13.096 min), 3,4′,5-trihydroxystilbene (Rt: 13.247 min), genkwanin (13.373 min), capsiate (Rt: 13.398 min), 6′-hydroxy-O-desmethylangolensin (Rt: 14.076 min) and 21-beta-hydroxyhederagenin (Rt: 15.08 min) identified in the QTOF-MS data of AEWP-sed, were absent in AEL-sed.

## 3. Discussion

In this study, the leaves and bark/root of *A. ebracteatus* were extracted with and without using a chlorophyll removal process, and the extracts were analyzed using UHPLC-QTOF-MS for their phytochemical profiles and their antioxidant, antimicrobial and anticancer properties. Free radicals are essential by-products generated during metabolic processes by the body. However, these radicals have the ability to form complexes through ionizing radiation, leading to oxidative stress, and they further attack biological molecules such as lipids, nucleic acid, and protein [10,11]. It has been widely acknowledged that reactive oxygen species and oxidative stress are extensively implicated in the pathophysiology of several diseases that plague humankind, including diabetes, cardiovascular diseases, and cancer [12,13]. The ability of a medicinal plant extract to exert any form of bioactivity is largely dependent on the phyto-constituents present in the plant. In addition, many natural products, including plant extracts or isolated bioactive compounds have displayed several pharmacological activities linked to their potential to modulate oxidative stress and exhibit antioxidant properties [13]. As such, the antioxidant activity of an extract plays a vital role in its pharmacological effects. In view of this, the antioxidant activities of *A. ebracteatus* extracts were evaluated using various established techniques, namely DPPH, ABTS, FRAP, MCA, and ORAC assays. Generally, the leaves extracts (AEL-sed and AEL-nor) exhibited the highest antioxidant properties. The ability of AEL-sed to scavenge DPPH (498.80 µmol TE/g) and ABTS (67.73 µmol TE/g) radicals, as well as reduce (FRAP: 1113.20 µmol TE/g) or chelate (MCA: 47.83 µmol EDTA/g) metal ions was of a greater extent compared to the extract from the stem/root. In a previous study, the ethanolic extract of *A. ebracteatus* was shown to exert DPPH (IC_50_: 0.12 ± 0.03 mg/L)-scavenging activity [9].

The results obtained from the total phenolics and flavonoids content indicated that AEL was rich in total phenolics (140.5 and 138.2  mg GAE/g for AEL-sed and AEL-nor, respectively) and flavonoids (110.4 and 107.6  mg CE/g for AEL-sed and AEL-nor, respectively). Earlier studies have indicated the presence of high levels of phenolics and flavonoids in *A. ebracteatus* [6]. The results obtained from our study confirmed the presence of phenolics and flavonoids. However, the TPC and TFC contents reported in our study were markedly higher [6]. The disparity in the phenolic and flavonoid contents may be attributed to the differences in plant origin, growth conditions, extraction methods, and the solvent employed for extraction.

Several bioactive molecules were detected in the *A. ebracteatus* extracts, including baicalin, apigenin, luteolin, glucocaffeic acid, caffeic acid, aesculin, diosmetin, genkwanin, saponarin, and hydrojuglone glucoside. These identified compounds could be responsible for the observed antioxidant properties since previous reports illustrated the antioxidant potential of these compounds through several mechanism in in vitro and in vivo models [12,14,15,16]. Prasansuklab and Tencomnao [6] reported the antioxidant potential of *A. ebracteatus* extract and suggested that its protective effects against oxidative stress injury were attributed to the presence of polyphenolic compounds in the extract (verbasoside, leucosceptoside A, isoverbascoside, and Vicenin-2). Furthermore, Ilori et al. [17] noted that polyphenolic compounds, such as verbascoside, leucosceptoside A, martynoside, β-hydroxyacteoside, pteleifoside G, magnolenin C, vecenin-2, shaftoside, luteolin-7-O-β-d glucuronide, and apigenin, which have several reported therapeutic activities such as antimicrobial, anticancer, wound healing, anti-inflammatory, anti-hair loss, and antioxidant properties, were reported to be present in *A. ebracteatus* [8,17].

The antimicrobial properties of the *A. ebracteatus* extracts could be obviously related to their high polyphenolic constituents. Pratoomsoot et al. [9] previously reported that extracts from *A. ebracteatus* showed significant antimicrobial activity against the *A. baumannii* DMST 10437, *E. coli* 4212, *S. aureus* DMST 8840, methicillin-resistant *S. aureus* DMST, *S. epidermidis* DMST 3547, *S. epidermidis* DMST 4343, and *S. pyogenes* DMST 30563 strains. The importance of controlling bacterial infections cannot be over-emphasized due to their prevailing negative effects in primary health care as well as the complications that arise from bacterial infections related to other diseases. An increasing number of reports illustrate the importance of medicinal plants in the treatment of bacterial infections [18]. The results indicated that *A. ebracteatus* extract showed significant antibacterial properties.

Cancer is a major cause of death globally and, unfortunately, there is no known cure for this dreaded disease [19]. As such, finding a cost-effective, alternate, and safer treatment for cancer is warranted. Medicinal plants have gained attention as alternative chemopreventive and therapeutic agents in recent years. In fact, numerous anticancer agents presently approved for cancer treatment or undergoing clinical trials as possible anticancer drugs have direct links to medicinal plants and are building blocks for the emergence of some synthetic anticancer agents [20]. Oral carcinogenesis is a multistep process that includes genetic events which lead to the disruption of the normal regulatory pathways that control cellular functions [21]. Oropharyngeal cancer and its treatment via chemotherapy causes several complications, including dysphagia, mucositis, pain, related infections, and bleeding [22]. Similar to chemotherapeutic agents, natural products such as phenethyl isothiocyanate [23], resveratrol [24], and curcumin [25] have been reported to have excellent anticancer efficacy with no or minimal side effects. The results from our study suggested that AEL-sed showed reasonable anticancer effects. Several phytochemicals, such as diosmetin, esculetin, isoacteoside, baicalin, isoamericanol A, luteolin, apigenin, and genkwanin, among several others identified in the extract, have been reported as promising anticancer agents in several in vitro and in vivo studies [26,27,28,29,30,31,32,33,34]. Therefore, *A. ebracteatus* extract contains several constituents with promising bioactivities that could be beneficial for the treatment of several disorders.

## 4. Materials and Methods

### 4.1. Chemicals and Reagents

Dimethyl sulfoxide and 3-(4,5-dimethylthiazol-2-yl)-2,5-diphenyltetrazolium bromide (MTT) were purchased from Sigma-Aldrich Corp., (St. Louis, MO, USA). Fetal bovine serum was procured from Biochrom GmbH (Berlin, Germany). RPMI-1640, phosphate buffer saline, and penicillin/streptomycin (U/mL) were purchased from PAA Laboratories GmbH (Pasching, Austria). 2′,7′-dichlorodihydrofluorescein diacetate, 0.25 % trypsin-EDTA, and stable L-glutamine were purchased from Gibco Life Technologies (Carlsbad, CA, USA). Phenotype oral squamous carcinoma cells (CLS-354/WT) were obtained from the Research Institute for Health Sciences, School of Allied Health Sciences, Walailak University, Nakhon Si Thammarat, Thailand. All other chemicals used were of analytical grade and used as purchased.

### 4.2. Plant Material

The leaves, stem and root of *A. ebracteatus* were collected from Surat Thani Province, Thailand. The plant was authenticated at the Faculty of Pharmaceutical Sciences, Prince of Songkla University, Thailand. The samples were powdered with a mechanical grinder (Jing Gongyi, JGY-800B, Yongkang, China) to fine particles and the powdered leaves and stem/roots were divided into two equal portions and extracted separately.

### 4.3. Preparation of A. ebracteatus Extracts

#### 4.3.1. Classical Ethanol Extraction

The powdered leaves (200 g) and stem/roots (200 g) were extracted with 2 L of 70% ethanol at a solvent/solid ratio of 10:1 (*v/w*) on a shaker for 24 h. Subsequently, the extraction mixture was filtered, and the resulting filtrate was dried under reduced pressure with a rotary evaporator at 45 °C. The dried extract of the leaves (AEL-nor) and back/root (AEWP-nor) were stored at 4 °C until further use.

#### 4.3.2. Extraction Using the Sedimentation Method

Likewise, 200 g of the powdered leaves and 200 g of the powdered stem/roots were subjected to 70% ethanol extraction at a solvent/solid ratio of 10:1 (*v/w*) on a shaker for 24 h. Thereafter, the solution obtained after filtration was concentrated to 30% of the initial volume, and the mixture was refrigerated at 4 °C for 24 h to sediment. Thereafter, the solution was decanted, and the top layer (without chlorophyll) was centrifuged (6000 rpm, 30 min at 4 °C). The supernatants obtained from the leaves extract (AEL-sed) and the stem/roots (AEWP-sed) were freeze-dried and stored until further use [35,36].

### 4.4. Total Phenolic and Flavonoid Content

The TPC and TFC of the extracts were determined based on previously reported protocol [37,38]. The TPC of the extracts was spectrophotometrically determined using the Folin–Ciocalteu method. Briefly, 0.1 mL of the extracts were added to 0.75 mL of 10% Folin–Ciocalteu reagent, and the mixture was allowed to stand for 5 min. Subsequently, 0.75 mL of a saturated solution of Na_2_CO_3_ was added, and the mixtures were incubated at room temperature for 3 h, while shaking randomly. Thereafter, the absorbance of the blue-colored solution was measured at 760 nm. TPC was expressed as mg gallic acid equivalent (GE)/g dry extract.

For the analysis of the TFC of the extracts, 800 µL of distilled water was mixed with 200 µL of the extract solution, 60 µL of 5% NaNO_2_ solution, and 60 µL of 10% AlCl_3_ solution. The mixture was allowed to stand for 5 min at room temperature and thereafter 400 µL of 1M NaOH solution was added. The mixture was made up to a volume of 2 mL with distilled water and thoroughly mixed. The absorbance of the solution was measured at 510 nm. TFC was calculated from the standard curve of catechin and expressed as mg catechin equivalent (CE)/g extract.

### 4.5. Antioxidant Activity

Measurements of the DPPH radical-scavenging activity (DPPH-RSA), ABTS radical-scavenging activity (ABTS-RSA), metal chelating activities (MCA), ferric reducing antioxidant power (FRAP), and oxygen radical antioxidant capacity (ORAC) of the extracts were performed using previously reported methods [37,38].

For ABTS-RSA, the stock solutions included 7.4 mM ABTS solution and 2.6 mM potassium persulfate solution. The working solution was prepared by mixing the two stock solutions in equal quantities. The mixture was allowed to react for 12 h at room temperature in the dark. The solution obtained (1 mL) was then diluted with 50 mL of distilled water to obtain an absorbance of 1.10 ± 0.02 units at 734 nm. The sample (150 μL) was mixed with 2850 μL of ABTS solution, and the mixture was left at room temperature for 1 h in the dark. The absorbance was then measured at 734 nm using a spectrophotometer. The blank was prepared in the same manner, except that distilled water was used instead of the sample. A standard curve of Trolox ranging from 50–600 μM was prepared. The activity was expressed as μmol Trolox equivalent (TE)/g solid.

The extracts sample (0.3 mL) was mixed with 2.7 mL of a methanolic solution containing DPPH (0.15 mM). The mixture was shaken vigorously and left to stand for 60 min in the dark (until stable absorption values were obtained) at room temperature (25 °C). The reduction of the DPPH-RSA was measured by continuously monitoring the decrease in absorbance at 517 nm. The DPPH scavenging activity was expressed as μmol Trolox equivalent (TE)/g solid.

The FRAP reagent was prepared by mixing acetate buffer (30 mM, pH 3.6) and 10 mM TPTZ solution in a 40 mM HCl and 20 mM iron (III) chloride solution in proportions of 10:1:1 (v/v). The sample solution (150 µL) was mixed with 2.85 mL of working FRAP reagent and incubated in dark conditions at room temperature for 30 min. The absorbance of the reaction mixture was read at 593 nm. The standard curve was prepared using Trolox ranging from 0–500 µM. The activity was expressed as µmol Trolox equivalent (TE)/g sample.

For MCA, 1 mL of extract was mixed with 3.7 mL of distilled water and the mixture was reacted with 0.1 mL of 2 mM FeCl_2_ and 0.2 mL of 5 mM ferrozine for 20 min. The absorbance was read at 532 nm. One milliliter of distilled water instead of the extract was used as a control. The chelating activity was expressed as μmol EDTA equivalent (EE)/g solid.

### 4.6. Antibacterial Activity

The minimum inhibitory concentration (MIC) and minimum bactericidal concentration (MBC) measurements of the extracts were performed against *Listeria monocytogenes* and *Escherichia coli* 0157, using the previously reported protocol [39].

### 4.7. Anticancer Efficacy Compounds

The anticancer efficacy of the extracts was tested against epithelium-like phenotype oral squamous carcinoma cell (CLS-354/WT) by an indirect method [40]. Briefly, carcinoma cells were cultured in RPMI-1640 supplemented with 10% fetal bovine serum, 1% *v/v* penicillin/streptomycin (U/mL), and 2mM stable ʟ-glutamine. Approximately, 1 × 10^4^ (cells/mL) cells were seeded in 96-well plates and incubated in an incubator with 5% CO_2_ at 37 °C. The cells were allowed to form a 70% confluent monolayer and treated with the extract (1600–12.5 μg/mL) and supplemented fresh RPMI-1640 as a negative control, in triplicate. The percentage of cell death was analyzed using MTT assay. The insoluble formazan crystals were solubilized with 99.9% DMSO, and the absorbance was measured at 560 nm using a multi-mode plate reader (BioTek, Winooski, VT, USA). The percentage of cell death was calculated.

### 4.8. Anti-Proliferative Effect of Extract

The in vitro scratch assay was evaluated to quantify the anti-migration capabilities of cells treated with the extracts. Briefly, CLS-354/WT cells were seeded at a cell density at 3 x 10^4^ cells/well in a 6-well plate. The confluent monolayer (70%) of the cells was scratched using a sterile pipette tip to create a wound of 1 mm width. Subsequently, the cells were washed with phosphate buffer (pH 7.4) to remove cellular debris and replaced with a fresh medium containing the extract above ED_50_ (50 % inhibition of cancer cell growth), or with RPMI-1640 medium as a negative control. Images of cell migration were captured at 0 and 24 h using a Carl Zeiss microscope Axio Vert. A1 (Konigsallee, Gottingen, Germany). The residual gap between the migrating cells was measured using Image J software (1.8.0_172).

### 4.9. UHPLC-ESI-QTOF-MS Profiling of the Extracts

The extracts (AEL-sed and AEWP-sed) with significant antioxidant and antimicrobial activities were selected for LCMS profiling. The experimental procedures and instrumental parameters were previously described by Eze and Tola [41]. The analysis was performed using an Agilent 1290 Infinity II LC System (Agilent Technologies, Santa Clara, CA, USA) equipped with an autosampler, a binary pump, a vacuum degasser, and a diode array detector. The extracts were separated on Agilent’s ZORBAX Eclipse Plus C18 column (150 × 2.1 mm, 1.8 µmm). The mobile phases consisted of (A) acidified Milli-Q water (0.1% formic acid) and (B) acetonitrile. The following parameters were employed for the elution: 0.50 min: 0% B; 16.50 min: 100% B; 17.50 min: 100% B; 20.00 min: 0.00% B; 22.00 min: 0.00% B; injection volume of 2.0 mL, flow rate of 0.2 µL min^−1^, and column temperature of 25 °C. The HPLC system was coupled to an Agilent 6545 LC/Q-TOF MS mass spectrometer equipped with a dual Agilent Jet Stream ESI negative mode, with a mass range of *m/z* 100 to 1500 at a scan rate of 1.00 spectrum per second. Accurate mass measurements by the instrument were ensured using an automated calibrant delivery system that continuously introduced a reference solution with a mass mix of *m/z* 112.985587 (TFA anion) to *m/z* 1033.988109 (HP-0921) in the ESI-negative mode, while a mass mix of *m/z* 121.050873 (purine) and *m/z* 922.009798 (HP-0921) were introduced in the ESI-positive mode. The parameters set for ESI-MS included: drying gas temperature: 325 °C; drying gas flow rate: 13 L minˉ1; nebulizer gas pressure: 35 psig; capillary voltage: 4000 V; fragmentor voltage: 175 V; radiofrequency voltage in the octupole: 750 V, and fixed collision energies of 10.00 eV, 20.00 eV and 40.00 eV. Data acquisition was performed on Mass Hunter Workstation Software Data Acquisition for Q-TOF, version B.08.00 (B8058.3 SP1) and QTOF Firmware, version 20.712.

### 4.10. Statistical Analysis

The results were expressed as mean ± standard error. Statistical analysis was determined by one-way ANOVA followed by Dunnett’s test using GraphPad Prism 6 (GraphPad Software Inc., La Jolla, CA, USA). Differences of *p* < 0.05 were considered significant.

## 5. Conclusions

In conclusion, the results from this study suggested that *A. ebracteatus* displayed promising antioxidant, antibacterial and anticancer activities. The leaves of the plant showed better activity in all the tested assays when compared to other extracts. Furthermore, UPLC-ESI-QTOF-MS analysis indicated that the plant is rich in polyphenolic compounds, including phenolic acids, flavonoids, iridoids, and o-glycosyl compounds. These results suggested that *A. ebracteatus* can be explored as a possible nutraceutical for the treatment of oxidative stress-related disorders. Further studies are needed to validate the in vivo pharmacological and activities, especially in unexplored and valuable aspects of *A. ebracteatus*.

## Figures and Tables

**Figure 1 molecules-27-01981-f001:**
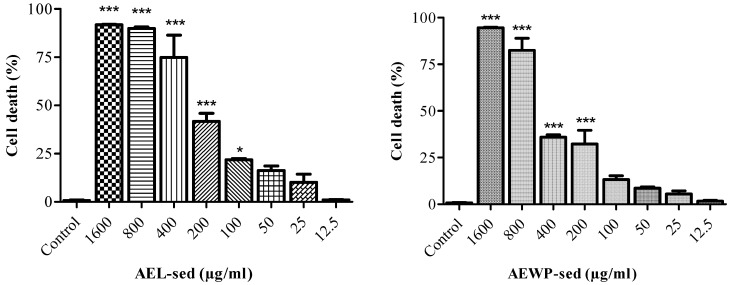
Epithelium-like phenotype oral squamous carcinoma cell (CLS-354/WT) death, in percent, treated with AEL-sed and AEWP-sed extracts (1600–12.5 μg/mL), using MTT assay. Data are expressed as mean ± SEM from at least three independent experiments and analyzed via one-way ANOVA with Dunnett’s test. * *p* < 0.05, *** *p* < 0.001 vs untreated control.

**Figure 2 molecules-27-01981-f002:**
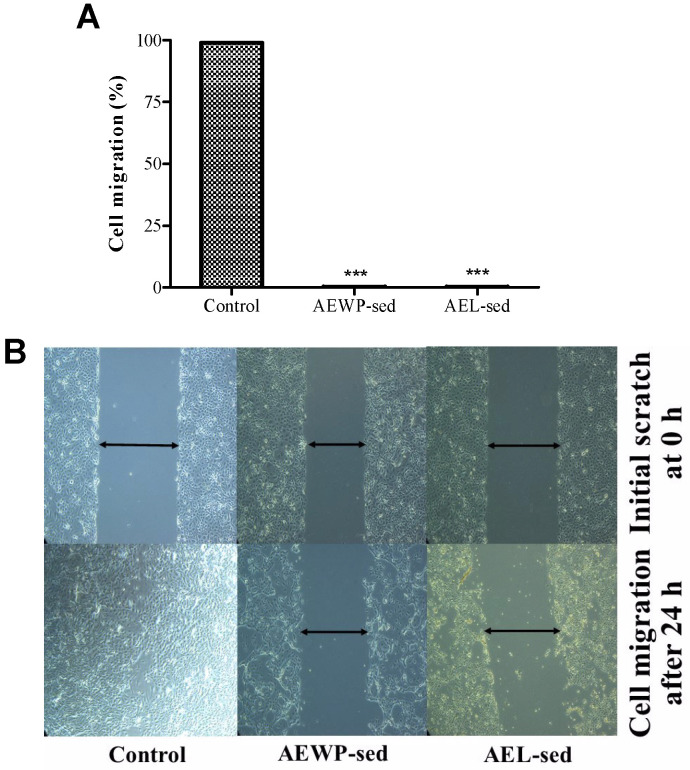
Antiproliferative effects of AEL-sed and AEWP-sed extracts (800 μg/mL) in epithelium-like phenotype oral squamous carcinoma cell (CLS-354/WT) at 0 and 24 h (**A**) Percentage migration calculated using the length of cell migration obtained from microscopic image (**B**) Microscopic photographs of CLS-354/WT cells migration by scratch technique. Data are expressed as mean ± SEM from at least 3 independent experiments and analyzed by one-way ANOVA followed by Dunnett’s test. *** *p* < 0.001 vs. untreated control.

**Figure 3 molecules-27-01981-f003:**
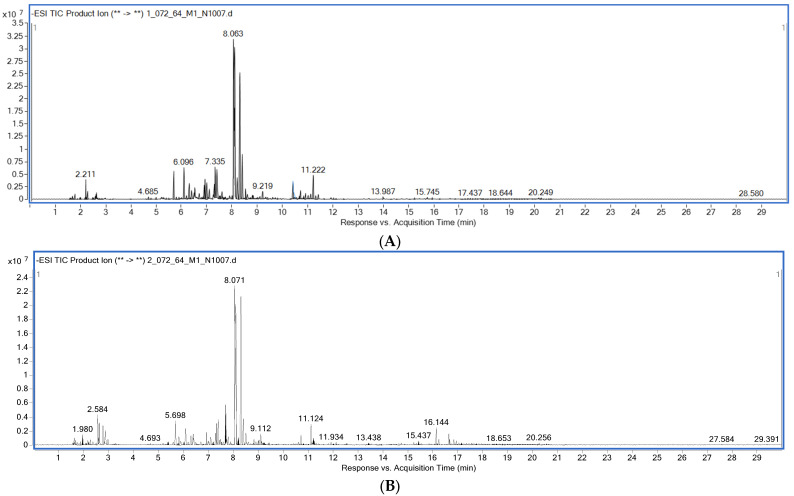
Total ion chromatograms of *A. ebracteatus* extract using UHPLC-ESI-QTOF-MS in the negative electrospray ionization mode showing the chromatogram intensity against the acquisition time; (**A**) AEL-sed, (**B**) AEWP-sed.

**Table 1 molecules-27-01981-t001:** Antioxidant activity of different extracts from *A. ebracteatus.*

Sample/Assay	AEWP-nor	AEWP-sed	AEL-nor	AEL-sed
TPC (mg GAE/g dry extract)	30.49 ± 0.10 ^e^	36.88 ± 0.10 ^d^	138.20 ± 0.10 ^b^	140.50 ± 0.10 ^a^
TFC (mg CE/g dry extract)	20.24 ± 0.20 ^e^	28.82 ± 0.10 ^d^	107.60 ± 0.02 ^b^	110.40 ± 0.50 ^a^
DPPH-RSA (µmol TE/g dry extract)	91.90 ± 0.40 ^f^	104.90 ± 0.08 ^e^	448.10 ± 1.20 ^b^	498.80 ± 0.40 ^a^
ABTS-RSA (µmol TE/g dry extract)	44.01 ± 0.10 ^d^	48.59 ± 0.08 ^c^	57.05 ±0.10 ^b^	67.73 ± 0.50 ^a^
FRAP (µmol TE/g dry extract)	182.80 ± 0.40 ^c^	223.01 ± 0.40 ^b^	1098.20 ± 7.1 ^a^	1113.20 ± 4.20 ^a^
MCA (µmol EDTA/g dry extract)	36.32 ± 0.10 ^e^	37.29 ± 0.05 ^d^	46.87 ± 0.20 ^b^	47.83 ± 0.01 ^a^
ORAC (µmol TE/g dry extract)	3.22 ± 0.20 ^b^	3.32 ± 0.60 ^b^	11.51 ± 0.50 ^a^	11.52 ± 0.30 ^a^

Different lowercase superscripts within the same column indicate significant difference (*p* < 0.05). Mean (*n* = 3); AEWP-nor: extract from the stem/root without dechlorophyllization; AEWP-sed: extract from the stem/root after the dechlorophyllization process using the sedimentation process; AEL-nor: extract from the leaves without dechlorophyllization; AEL-sed: extract from the leaves after the dechlorophyllization process using the sedimentation process.

**Table 2 molecules-27-01981-t002:** Antibacterial activity of different extracts from *A. ebracteatus.*

Samples	MIC		MBC	
	EC	LM	EC	LM
AEWP-nor	1.00 ^a^	1.00 ^a^	2.00 ^a^	2.00 ^a^
AEWP-sed	1.00 ^a^	1.00 ^a^	2.00 ^a^	2.00 ^a^
AEL-nor	0.25 ^c^	0.50 ^b^	0.50 ^c^	1.00 ^b^
AEL-sed	0.25 ^c^	0.50 ^b^	0.50 ^c^	1.00 ^b^

EC, *Escherichia coli*; LM, *Listeria monocytogenes*; MIC, minimum inhibitory concentration; MBC, minimum bactericidal concentration. Different lowercase superscripts within the same column indicate significant difference (*p* < 0.05). Mean (*n* = 3).

**Table 3 molecules-27-01981-t003:** Compounds tentatively identified in AEL-sed using UHPLC-ESI-QTOF-MS analysis.

No	Rt (min)	Accurate Mass (*m/z*)	Calculated Mass (Da)	Score (DB)	Predicted Formula	Compound Identity
1	2.997	337.0775	338.0848	98.14	C_12_H_18_O_11_	L-Ascorbic acid-2-glucoside
2	3.311	225.0015	226.009	63.57	C_8_H_7_ClN_4_S	6-(2-Chloroallylthio)purine
3	4.076	134.0471	135.0544	87.88	C_5_H_5_N_5_	Adenine
4	4.604	330.119	331.1262	98.71	C_14_H_21_NO_8_	5′-O-beta-D-Glucosylpyridoxine
5	4.729	128.0351	129.0426	95.05	C_5_H_7_NO_3_	(R)-(+)-2-Pyrrolidone-5-carboxylic acid
6	4.767	243.0624	244.0697	99.79	C_9_H_12_N2O_6_	Pseudouridine
7	4.805	174.077	175.0843	99.74	C_7_H_13_NO_4_	Calystegine B5
8	4.854	180.0664	181.0736	99.62	C_9_H_11_NO_3_	3-Amino-3-(4-hydroxyphenyl)propanoate
9	5.005	282.0842	283.0915	98.4	C_10_H_13_N_5_O_5_	Guanosine
10	5.055	150.042	151.0492	98.18	C_5_H_5_N_5_O	8-Hydroxyadenine
11	5.256	405.1395	406.1468	98.28	C_17_ H_26_O_11_	Morroniside
12	5.269	477.1608	478.1681	97.79	C_20_H_30_O_13_	Kelampayoside A
13	5.356	108.0456	109.0529	98.09	C_6_H_7_NO	3-Hydroxy-2-Methylpyridine
14	5.52	371.0979	372.1052	98.94	C_16_H_20_O_10_	Dihydroferulic acid 4-O-glucuronide
15	5.532	331.0668	332.0741	99.44	C_13_H_16_O_10_	4-Glucogallic acid
16	5.959	355.1044	356.1116	95.68	C_16_H_20_O_9_	1-O-2′-Hydroxy-4′-methoxycinnamoyl-b-D-glucose
17	6.16	461.1665	462.1738	99.05	C_20_H_30_O_12_	Verbasoside
18	6.26	403.1245	404.1318	96.86	C_17_H_24_O_11_	Oleoside 11-methyl ester
19	6.336	341.0875	342.0947	83.78	C_15_H_18_O_9_	Glucocaffeic acid
20	6.36	179.0349	180.0423	96.04	C_9_H_8_O_4_	Caffeic Acid
21	6.637	339.0721	340.0794	99.12	C_15_H_16_O_9_	Aesculin
22	7.038	353.088	354.0953	99.19	C_16_H_18_O_9_	Chlorogenic acid
23	7.114	593.151	594.1583	98.71	C_27_H_30_O_15_	Saponarin
24	7.164	387.1657	388.1729	97.14	C_18_H_28_O_9_	2-[4-(3-Hydroxypropyl)-2-methoxyphenoxy]-1,3-propanediol-1-xyloside
25	7.39	639.1929	640.2024	57.65	C_29_H_36_O_16_	beta-Hydroxyacteoside
26	7.415	637.1046	638.1119	99.11	C_27_H_26_O_18_	Scutellarein-7-glucuronosyl-(1->2)-glucuronide
27	7.54	563.1406	564.1476	97.71	C_26_H_28_O_14_	Luteolin 3′-methyl ether 7,4′-dixyloside
28	7.565	415.1608	416.1682	95.16	C_19_H_28_O_10_	Phenylethyl primeveroside
29	7.817	177.0195	178.0267	86.68	C_9_H_6_O_4_	Esculetin
30	7.842	399.166	400.1734	95.33	C_19_H_28_O_9_	Corchoionoside B
31	7.917	621.11	622.117	98.15	C_27_H_26_O_17_	Genistein 4′,7-O-diglucuronide
32	7.942	383.0622	384.0693	97.99	C_16_H_16_O_11_	2-O-Feruloylhydroxycitric acid
33	7.967	353.0517	354.0588	98.05	C_15_H_14_O_10_	2-O-p-Coumaroylhydroxycitric acid
34	8.018	337.0932	338.1004	98.79	C_16_H_18_O_8_	Hydrojuglone glucoside
35	8.369	461.0725	462.0797	98.96	C_21_H_18_O_12_	3-Methylellagic acid 8-rhamnoside
36	8.432	623.1996	624.2065	96.46	C_29_H_36_O_15_	Isoacteoside
37	8.62	637.2143	638.2212	98.89	C_30_H_38_O_15_	4′-Hydroxy-5,7,2′-trimethoxyflavanone-4′-rhamnosyl-(1->6)-glucoside
38	8.745	429.1764	430.1837	97.61	C_20_H_30_O_10_	Phenethyl rutinoside
39	8.871	579.1728	580.1799	97.92	C_27_H_32_O_14_	Cascaroside F
40	8.884	445.0774	446.0847	97.07	C_21_H_18_O_11_	Baicalin
41	8.971	475.088	476.0953	98.09	C_22_H_20_O_12_	Diosmetin 7-O-beta-D-glucuronopyranoside
42	9.147	665.2081	666.2159	88.16	C_31_H_38_O_16_	Tetramethylquercetin 3-rutinoside
43	9.273	651.2289	652.2359	97.31	C_31_H_40_O_15_	(-)-Matairesinol-4′-[apiosyl-(1->2)-glucoside]
44	9.373	413.2173	414.2247	95.11	C_21_H_34_O_8_	(4R,5S,7R,11S)-11,12-Dihydroxy-1(10)-spirovetiven-2-one 11-glucoside
45	9.386	433.1498	434.1571	96.09	C_22_H_26_O_9_	Vestitone 7-glucoside
46	9.624	431.1345	432.1419	95.75	C_22_H_24_O_9_	4′-O-Methylglucoliquiritigenin
47	9.8	591.2074	592.2147	96.83	C_29_H_36_O_13_	Osmanthuside B
48	10.151	503.1552	504.1625	72.14	C_25_H_28_O_11_	Sergeolide
49	10.277	473.1438	474.1511	92.54	C_24_H_26_O_10_	Luteolin 7,3′-dimethyl ether 5-glucoside
50	10.327	275.092	276.0993	98.16	C_15_H_16_O_5_	5-De-O-methyltoddanol
51	10.478	285.0405	286.0478	98.52	C_15_H_10_O_6_	Luteolin
52	10.955	329.1037	330.1107	85.17	C_18_H_18_O6	Isoamericanol A
53	11.03	207.0666	208.0738	97.72	C_11_H_12_O_4_	5-(3′,5′-Dihydroxyphenyl)-gamma valerolactone
54	11.18	220.0613	221.0685	99.48	C_11_H_11_NO_4_	Methyl dioxindole-3-acetate
55	11.193	329.2332	330.2404	98.67	C_18_ H_34_O_5_	9S,10S,11R-trihydroxy-12Z-octadecenoic acid
56	11.281	269.0455	270.0528	98.69	C_15_H_10_O_5_	Apigenin
57	11.432	299.0563	300.0635	98.53	C_16_ H_12_O_6_	Diosmetin
58	11.482	619.1446	620.1518	96.93	C_32_H_28_O_13_	Apigenin-7-(3″-acetyl-6″-E-p-coumaroylglucoside)
59	11.934	268.0611	269.0682	97.25	C_15_H_11_NO_4_	Evoxanthidine
60	13.088	307.1907	308.198	81.32	C_18_H_28_O_4_	Dihydrocapsiate
61	13.289	675.358	676.3652	94.84	C_33_H_56_O_14_	Gingerglycolipid A
62	13.44	293.1752	294.1825	98.16	C_17_H_26_O_4_	Myrsinone
63	14.067	273.0765	274.0837	98.17	C_15_ H_14_O_5_	2,3,4-Trihydroxy-4′-ethoxybenzophenone
64	14.545	241.0866	242.0938	98.73	C_15_H_14_O_3_	Resveratrol 4′-methyl Ether
65	15.8	291.0425	292.0498	98.05	C_15_H_13_ClO_4_	Chlorosesamone

**Table 4 molecules-27-01981-t004:** Compounds tentatively identified in AEWP-sed using UHPLC-ESI-QTOF-MS analysis.

No	Rt (min)	Accurate Mass (*m/z*)	Calculated Mass (Da)	Score (DB)	Predicted Formula	Compound Identity
1	2.251	629.1697	630.1768	77.85	C_27_H_34_O_17_	Leucodelphinidin-3-O-(beta-D-glucopyranosyl-(1->4)-alpha-L-rhamnopyranoside)
2	2.302	191.0562	192.0635	99.67	C_7_H_12_O_6_	Quinic acid
3	2.427	827.2658	828.2731	97.02	C_30_H_52_O_26_	Verbascose
4	2.528	503.1612	504.1685	98.71	C_18_H_32_ O_16_	Nephritogenoside
5	2.653	683.225	684.2323	80.52	C_37_H_36_N_2_O_11_	Citbismine C
6	2.654	341.1091	342.1163	98.89	C_12_ H_22_ O_11_	2-O-a-D-Galactopyranuronosyl-L-rhamnose
7	3.156	290.0878	291.095	99.06	C_11_H_17_NO_8_	Sarmentosin epoxide
8	3.193	665.2136	666.2208	97.88	C_24_H_42_O_21_	Fagopyritol A3
9	3.331	225.0016	226.009	65.47	C_8_H_7_ClN_4_S	6-(2-Chloroallylthio)purine
10	3.381	203.0196	204.0269	99.89	C_7_H_8_O_7_	Daucic acid
11	4.762	243.0623	244.0697	99.5	C_9_H_12_N_2_O_6_	Pseudouridine
12	4.837	174.077	175.0843	99.62	C_7_H_13_NO_4_	Calystegine B5
13	5.038	282.0839	283.0913	98.71	C_10_H_13_N_5_O_5_	Guanosine
14	5.264	477.1609	478.1681	98.24	C_20_H_30_O_13_	Kelampayoside A
15	5.904	329.0876	330.0948	84.7	C_14_H_18_O_9_	3′-Glucosyl-2′,4′,6′-trihydroxyacetophenone
16	6.093	359.0981	360.1055	97.46	C_15_H_20_O_10_	6′-Methoxypolygoacetophenoside
17	6.168	461.1666	462.1738	99.21	C_20_H_30_O_12_	Verbasoside
18	6.243	167.0348	168.0421	99.62	C_8_H_8_O_4_	Dihydroxyphenylacetic acid
19	6.268	343.1027	344.1102	97.17	C_15_H_20_O_9_	4′,6′-Dihydroxy-2′-methoxyacetophenone 6′-glucoside
20	6.344	341.0875	342.0947	82.32	C_15_H_18_O_9_	Glucocaffeic acid
21	6.368	403.1241	404.1315	97.32	C_17_H_24_O_11_	Oleoside 11-methyl ester
22	6.469	179.035	180.0416	60.28	C_9_H_8_O_4_	Caffeic Acid
23	6.871	513.2184	514.2256	98.64	C_21_H_38_O_14_	2-O-(beta-D-galactopyranosyl-(1->6)-beta-D-galactopyranosyl) 2S,3R-dihydroxynonanoic acid
24	6.921	431.1556	432.1629	99.13	C_19_H_28_O_11_	Benzyl gentiobioside
25	6.971	387.1293	388.1365	98.02	C_17_H_24_O_10_	Geniposide
26	6.984	326.0886	327.0955	94.43	C_14_H_17_NO_8_	Blepharin
27	7.047	457.1356	458.1427	98.51	C_20_H_26_O_12_	7-Hydroxy-4-methylphthalide O-[arabinosyl-(1->6)-glucoside]
28	7.122	593.1515	594.1587	98.85	C_27_H_30_O_15_	Saponarin
29	7.222	293.124	294.1313	99.71	C_12_H_22_O_8_	Ethyl 3-O-beta-D-glucopyranosyl-butanoate
30	7.272	785.2499	786.257	97.39	C_35_H_46_O_20_	Echinacoside
31	7.461	639.193	640.2006	93.47	C_29_H_36_O_16_	beta-Hydroxyacteoside
32	7.511	563.1404	564.1476	97.59	C_26_H_28_O_14_	Luteolin 3′-methyl ether 7,4′-dixyloside
33	7.524	137.0247	138.0319	99.35	C_7_H_6_O_3_	2,5-Dihydroxybenzaldehyde
34	7.649	327.1085	328.1158	99.26	C_15_H_20_O_8_	Dihydromelilotoside
35	7.787	581.224	582.2314	97.41	C_28_H_38_O_13_	(+)-Lyoniresinol 9-glucoside
36	7.925	383.0619	384.0691	99.1	C_16_H_16_O_11_	2-O-Feruloylhydroxycitric acid
37	7.95	621.1096	622.1166	98.47	C_27_H_26_O_17_	Genistein 4′,7-O-diglucuronide
38	7.975	353.0514	354.0586	99.36	C_15_H_14_O_10_	2-O-p-Coumaroylhydroxycitric acid
39	8.051	337.0925	338.0999	99.16	C_16_H_18_O_8_	Hydrojuglone glucoside
40	8.377	551.2125	552.2198	97.21	C_27_ H_36_O_12_	Prupaside
41	8.427	623.1992	624.2061	98.06	C_29_H_36_O_15_	Isoacteoside
42	8.628	637.2133	638.2204	97.93	C_30_H_38_O_15_	4′-Hydroxy-5,7,2′-trimethoxyflavanone 4′-rhamnosyl-(1->6)-glucoside
43	9.055	563.1037	564.1109	98.35	C_25_H_24_O_15_	Larycitrin 3-(4″-malonylrhamnoside)
44	9.18	533.093	534.1001	95.67	C_24_H_22_O_14_	2′-Hydroxygenistein 7-(6″-malonylglucoside)
45	9.256	665.2085	666.2156	98.23	C_31_H_38_O_16_	Tetramethylquercetin 3-rutinoside
46	9.381	503.0817	504.089	92.63	C_23_H_20_O_13_	Gomphrenol 3-methylether 4′-glucuronide
47	9.557	393.1545	394.1624	88.04	C_20_H_26_O_8_	Gibberellin A43
48	9.633	431.134	432.1416	86.12	C_22_H_24_O_9_	4′-O-Methylglucoliquiritigenin
49	9.833	144.0455	145.0528	87.81	C_9_H_7_NO	4-formyl Indole
50	9.934	291.0871	292.0943	85.16	C_15_H_16_O_6_	trans-Grandmarin
51	10.197	395.2066	396.214	94.7	C_21_H_32_O_7_	Isopetasoside
52	10.335	275.0919	276.0993	97.91	C_15_H_16_O_5_	5-De-O-methyltoddanol
53	10.36	213.0917	214.099	98.96	C_14_H_14_O_2_	Ethyl 1-naphthylacetic acid
54	10.436	285.0403	286.0476	99.79	C_15_H_10_O_6_	Luteolin
55	10.586	721.2332	722.2405	43.96	C_34_ H_42_O_17_	Amorphigenol O-vicianoside
56	10.687	135.0815	136.0889	94.23	C_9_H_12_O	2-(1-Pentenyl)furan
57	11.038	207.0662	208.0734	99.66	C_11_H_12_O_4_	5-(3′,5′-Dihydroxyphenyl)-gamma-valerolactone
58	11.314	269.0454	270.0527	98.57	C_15_H_10_O_5_	Apigenin
59	11.402	299.0558	300.0631	97.59	C_16_H_12_O_6_	Diosmetin
60	11.515	619.1449	620.1521	96.73	C_32_H_28_O_13_	Apigenin 7-(3″-acetyl-6″-E-p-coumaroylglucoside)
61	12.569	375.1447	376.1519	83.09	C_20_H_24_O_7_	alpha-Peroxyachifolide
62	12.946	223.1338	224.1411	99.62	C_13_H_20_O_3_	Methyl jasmonate
63	13.096	381.0973	382.1046	97.02	C_21_H_18_O_7_	Mollicellin B
64	13.247	227.0709	228.0782	98.97	C_14_H_12_O_3_	3,4′,5-Trihydroxystilbene
65	13.373	283.0606	284.0679	84.78	C_16_ H_12_O_5_	Genkwanin
66	13.398	305.1752	306.1825	84.31	C_18_H_26_O_4_	Capsiate
67	13.448	309.2069	310.2141	84.83	C_18_H_30_O_4_	Auxin b
68	13.448	293.1755	294.1828	95.67	C_17_H_26_O_4_	Myrsinone
69	14.076	273.0764	274.0838	96.81	C_15_H_14_O_5_	6′-Hydroxy-O-desmethylangolensin
70	14.565	307.1912	308.1984	84.43	C_18_H_28_O_4_	Dihydrocapsiate
71	15.08	487.3423	488.3494	95.88	C_30_H_48_O_5_	21beta-Hydroxyhederagenin
72	15.959	423.1805	424.1877	97.88	C_25_H_28_O_6_	1,7-Dihydroxy-3,6-dimethoxy-2,8-diprenylxanthone
73	16.737	407.1858	408.193	98.62	C_25_H_28_O_5_	1-Hydroxy-3,5-dimethoxy-2,4-diprenylxanthone

## Data Availability

Not applicable.

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
