# Peer review of "New Insights on Acanthus ebracteatus Vahl: UPLC-ESI-QTOF-MS Profile, Antioxidant, Antimicrobial and Anticancer Activities"

_molecules, 2022, doi:10.3390/molecules27061981_

Round 1

Reviewer 1 Report

In the presented manuscript authors characterized phytochemical and pharmaclogical profiles of the leaves and stem/root extracts of Acanthus ebracteatus. Manuscript is interesting and worth to publish. However, I have a few comments and remarks.

Line 68: Add the space between text and bracket the same In line 185.

Line 71-75: The units for individual assays should be included.

Table 1. The description „Antioxidant activity” is to short and non-exhaustive.

Table 1. Check the signed differences between means: for ex ample for FRAP 223.01 ± 0.4 „bc” is not for sense. 

Table 2. Standard deviations or standard errors for data In Table 2 should be included.

Table 2. The title of table should be more exhaustive and comma should be replaced by “and”

Lines 83-83: Use the Italic style for the Latin names.

Line 253: How the leaves and stem/roots were powdered?

The quality of Fig. 3 is poor.

Author Response

In the presented manuscript authors characterized phytochemical and pharmacological profiles of the leaves and stem/root extracts of Acanthus ebracteatus. Manuscript is interesting and worth to publish. However, I have a few comments and remarks.

Line 68: Add the space between text and bracket the same In line 185.

Response: We have added the space as suggested

Line 71-75: The units for individual assays should be included.

Response: We have added the units of individual assays as suggested

Table 1. The description „Antioxidant activity” is to short and non-exhaustive.

Response: Thank you for your comment. The detailed description of antioxidant assays has been included in the revised text.

Table 1. Check the signed differences between means: for ex ample for FRAP 223.01 ± 0.4 „bc” is not for sense.

Response: We confirm that the differences between means is absolutely correct

Table 2. Standard deviations or standard errors for data In Table 2 should be included.

Response: We appreciate your comment. Ideally when antibacterial properties is represented in the form of minimum inhibitory concentration (MIC), they are not presented with standard error or deviation unless it is presented as zones of inhibition. This is the standard practice. Please see several other articles representing antimicrobial activities with MIC. The result are means of triplicate findings.

  • Synergistic antibacterial effects of meropenem in combination with aminoglycosides against carbapenem-resistant Escherichia coli Harboring blaNDM-1 and blaNDM-5. Antibiotics, 2021, 10, 1023
  • Evaluation of the synergistic antibacterial effects of fosfomycin in combination with selected antibiotics against carbapenem–resistant Acinetobacter baumannii. Pharmaceuticals, 2021, 14, 185.
  • Antidiabetic, antioxidant and antimicrobial activity of the aerial part of Tiliacora triandra. South African Journal of Botany, 2019, 125, 337–343

Table 2. The title of table should be more exhaustive and comma should be replaced by “and”

Response: The title of Table 2 has been revised  

Lines 83-83: Use the Italic style for the Latin names.

Response: All Latin names are in italics form as suggested

Line 253: How the leaves and stem/roots were powdered?

Response: It was powdered with a grinder and it has been included in the revised

The quality of Fig. 3 is poor.

Response: The image quality has been improved

Reviewer 2 Report

Dear authors,

The manuscript entitled "New insights on Acanthus ebracteatus Vahl: UPLC-ESI-QTOF-MS profile, antioxidant, antimicrobial and anticancer activities" concerns interesting research material with great potential, but I have some comments and observations.

  1. Please pay attention to the correct formatting of the captions above the tables and the correct formatting of the tables, eg tables 1 and 2. Please also include in the table descriptions which units of m / v and antioxidant activity have been used. This will increase the readability of the results.
  2.  "described in material and method Section 4.2, were examined using several in vitro assays (DPPH-RSA, ABTS-RSA, FRAP, MCA and ORAC)." 66-67 Since the materials and methods are given later in the text and not before the results, acronyms should be clarified beforehand to facilitate data analysis.
  3.  "The effects of the extracts on cellular apoptosis of CLS-354 / WT was testd" line 91 - apoptosis cannot be detected with MTT alone - this test verifies cell viability and not the type of cell death. This sentence should be revised.
  4. "inhibited the proliferation of cells with cellular apoptosis." line 104. Again, how was apoptosis found?
  5. My main concern is the overly laconic description of the LC-MS methodology and the results. The protocol and the apparatus used (HPLC model, detector types and manufacturers, solvent types, gradient type) should be given in greater detail. Determining the compounds only on the route of molecular masses entails a high percentage of error. Why are the absorbance maxima of the compounds not reported? "Likewise, the retention times, absorbance spectra and the data of MS were used to" line 139. Authors should report the literature data on the retention times / substance standards / databases on which they relied in identifying probable substances.
  6. "The total phenolic content (TPC), total flavonoid content (TFC), DPPH" line 271 Total phenolic and flavonoid content is not antioxidant activity. The protocols used to analyze these characteristics should be briefly described here, rather than citing only the literature.
    Best regards,

Author Response

The manuscript entitled "New insights on Acanthus ebracteatus Vahl: UPLC-ESI-QTOF-MS profile, antioxidant, antimicrobial and anticancer activities" concerns interesting research material with great potential, but I have some comments and observations.

Please pay attention to the correct formatting of the captions above the tables and the correct formatting of the tables, eg tables 1 and 2. Please also include in the table descriptions which units of m / v and antioxidant activity have been used. This will increase the readability of the results.

Response: The captions, title and the description on the tables have been done to conform to the standard of the journal and also in similar fashion as previously published articles in the journal

 "described in material and method Section 4.2, were examined using several in vitro assays (DPPH-RSA, ABTS-RSA, FRAP, MCA and ORAC)." 66-67 Since the materials and methods are given later in the text and not before the results, acronyms should be clarified beforehand to facilitate data analysis.

Response: As suggested all acronyms have been clearly defined at the first place where it is measured in the manuscript

The effects of the extracts on cellular apoptosis of CLS-354 / WT was tested" line 91 - apoptosis cannot be detected with MTT alone - this test verifies cell viability and not the type of cell death. This sentence should be revised.

"inhibited the proliferation of cells with cellular apoptosis." line 104. Again, how was apoptosis found?

Response: These statement have been revised in the revised manuscript

My main concern is the overly laconic description of the LC-MS methodology and the results. The protocol and the apparatus used (HPLC model, detector types and manufacturers, solvent types, gradient type) should be given in greater detail. Determining the compounds only on the route of molecular masses entails a high percentage of error. Why are the absorbance maxima of the compounds not reported? "Likewise, the retention times, absorbance spectra and the data of MS were used to" line 139. Authors should report the literature data on the retention times / substance standards / databases on which they relied in identifying probable substances.

Response: As suggested, we have added more details about the methodology and apparatus used for the LC-MS analysis. As regards the report of the compounds, we did report based on the how previous reports have conducted their reports.

  • Ethanolic guava leaf extracts with different chlorophyll removal processes: Anti-melanosis, antibacterial properties and the impact on qualities of Pacific white shrimp during refrigerated storage. Food Chemistry, 2021, 341, 128251
  • Phytochemical and in silico ADME/Tox analysis of Eruca sativa extract with antioxidant, antibacterial and anticancer potential against Caco-2 and HCT-116 colorectal carcinoma cell lines. Molecules, 2022, 27, 1409.
  • Antioxidant, antimicrobial activities and characterization of polyphenol-enriched extract of Egyptian celery (Apium graveolens L., Apiaceae) aerial parts via UPLC/ESI/TOF-MS. Molecules, 2022, 27, 698.
  • LC-MS/MS screening, total phenolic, flavonoid and antioxidant contents of crude extracts from three Asclepiadaceae species growing in Jordan. Molecules, 2022, 27, 859
  • Secondary metabolites profiling, biological activities and computational studies of Abutilon figarianum Webb (Malvaceae). Processes, 2020, 8, 336.
  • Chromolaena odorata (Siam weed): A natural reservoir of bioactive compounds with potent anti-fibrillogenic, antioxidative, and cytocompatible properties. Biomedicine & Pharmacotherapy, 2021, 141, 111811.
  • Investigation into the biological properties, secondary metabolites composition, and toxicity of aerial and root parts of Capparis spinosa L.: An important medicinal food plant. Food and Chemical Toxicology, 2021, 155, 112404.

In addition, we cannot include or report literature data on the retention times of all the compound identified in this study as it is obviously known that differences in the methods, solvents used and even the apparatus can affect the retention time of compounds significantly. In addition, several of these compound do not even have reported retention time in the literature .

"The total phenolic content (TPC), total flavonoid content (TFC), DPPH" line 271 Total phenolic and flavonoid content is not antioxidant activity. The protocols used to analyze these characteristics should be briefly described here, rather than citing only the literature.

Response: The protocols used for the analysis of TPC and TFC has been briefly described in the revised manuscript

Reviewer 3 Report

molecules-1636711

The manuscript gives detailed experimental information supporting the medicinal properties of plants specifically Sea Holly. The authors have evaluated the chemical composition of A. ebracteatus leaves and stem/root extracts, and biological studies revealed promising antioxidant, antibacterial and cytotoxic properties. Fairly comprehensive experimental, and biological studies are solid. This present manuscript thus fits well into the Molecules readership's interests and may become suitable for publication after the revision. However, the issues need to resolve before the possible publication of these results.

  1. The abstract section can be simplified as there are a lot of keywords not defined properly which might suitable for common readers or a large audience of journals.
  2. In the result and discussion section, DPPH-RSA, ABTS-RSA, FRAP, ORAC, MTT, ED50, etc needs to be defined in first place in the manuscript before using their standard accepted abbreviations.
  3. The author should provide a better resolution of figure 3 as in the current figure, the ESI-MS values are not clear and also values should be mentioned in the figure caption.
  4. Line 184, the IC50 values should be defined and 50 as a subscript.
  5. The author should comment on the nature of free radical generating during the course of reactions.

Author Response

The manuscript gives detailed experimental information supporting the medicinal properties of plants specifically Sea Holly. The authors have evaluated the chemical composition of A. ebracteatus leaves and stem/root extracts, and biological studies revealed promising antioxidant, antibacterial and cytotoxic properties. Fairly comprehensive experimental, and biological studies are solid. This present manuscript thus fits well into the Molecules readership's interests and may become suitable for publication after the revision. However, the issues need to resolve before the possible publication of these results.

The abstract section can be simplified as there are a lot of keywords not defined properly which might suitable for common readers or a large audience of journals.

Response: We have simplified the abstract as much as possible

In the result and discussion section, DPPH-RSA, ABTS-RSA, FRAP, ORAC, MTT, ED50, etc needs to be defined in first place in the manuscript before using their standard accepted abbreviations.

Response: All abbreviation have been properly defined at the first place they were mentioned in the revised manuscript

The author should provide a better resolution of figure 3 as in the current figure, the ESI-MS values are not clear and also values should be mentioned in the figure caption.

Response: We have provided a clearer figure 3 in the revised manuscript

Line 184, the IC50 values should be defined and 50 as a subscript.

Response: We have corrected as a subscript

The author should comment on the nature of free radical generating during the course of reactions.

Response: This has been included in the revised manuscript

Round 2

Reviewer 1 Report

The authors corrected most of my comments accordingly. However, I have still two comments

Firstly: Line 269: Authors answered my first comments and have written that “The samples were powdered with a mechanical grinder” It is not enough to answer. The type of grinder should be included and most importantly, the size of particles should be included too, because the size of particles decides about the extraction process.

Secondly, the Authors stated that FRAP 223.01 ± 0.4 „bc” is correct. What is the sense to give “bc” together in this line, “b” or “c” should be included? Please explain. Please, also explain under the tables a,b,c. In my opinion, it shows differences between means in lines not in the rows.

Author Response

The authors corrected most of my comments accordingly. However, I have still two comments

Firstly: Line 269: Authors answered my first comments and have written that “The samples were powdered with a mechanical grinder” It is not enough to answer. The type of grinder should be included and most importantly, the size of particles should be included too, because the size of particles decides about the extraction process.

Response: The name of the grinder and the size of the particles as been included in the manuscript

Secondly, the Authors stated that FRAP 223.01 ± 0.4 „bc” is correct. What is the sense to give “bc” together in this line, “b” or “c” should be included? Please explain. Please, also explain under the tables a,b,c. In my opinion, it shows differences between means in lines not in the rows.

Response: We thank the reviewer for the comment, we have clearly stated what the symbols represents

Reviewer 2 Report

Dear authors,

Thank you for your reply to the review and for the corrections made.
Yours faithfully,

Author Response

We thank the reviewer for the comment